# SNP associations in the L-citrulline metabolic pathway and vascular aging in the Japanese population

Dai Nogimura[1], Kazuki Moriyasu[1], Sachiko Ishida[2], Masakazu Kohda[2], Takayuki Yazawa[2], Masahiko Morita[1]*

1 Institute of Health Sciences, Kirin Holdings Company, Ltd., Fujisawa, Kanagawa, Japan, 2 DeNA Life Science, Inc., R&D Group, Tokyo, Japan

* masahiko_morita@kirin.co.jp

## Abstract

Decreased nitric oxide (NO) production from the vascular endothelium is a major factor for vascular aging. Because vascular aging has few specific subjective symptoms, assessing the susceptibility to vascular aging is beneficial for its early detection and improvement. Therefore, this study evaluated the associations between single nucleotide polymorphisms (SNPs) on L-citrulline (Cit) pathways essential for NO production and the health characteristics involved in vascular aging using a candidate gene approach. Associations with a significance level included those between the *KCNMB4* rs17108108 C allele and tendency to gain weight, the *ADCY8* rs6470860 G allele and numbness of limbs, the *NOS1* rs2271987 T allele and lower back pain, and the *PDE9A* rs2284972 G allele and body pain with negative mood states. A genome-wide association study was also conducted to analyze SNPs more extensively across genes related to Cit and NO metabolism, which revealed genome-wide significant associations between the *PRMT6* rs12028323 C allele and mood disturbance. These significant associations could be explained by the change in downstream NO signaling, supporting the relationship between the investigated traits and vascular function. These traits can be manifested as subjective symptoms of vascular aging. Therefore, the identified SNPs could predict susceptibility to the subjective symptoms of vascular aging, which could lead to genotype-based personalized interventions of Cit for an efficient improvement of vascular health.

## Introduction

Cardiovascular conditions are closely associated with aging as William Osler stated "A man is as old as his arteries." Vascular aging is a phenomenon caused by age-related stiffening and thickening of vascular wall that results in the loss of vascular flexibility and expandability. A major factor for vascular aging is decreased nitric oxide (NO) production from the vascular endothelium. Age-related decline in

**Data availability statement:** All relevant data are within the manuscript and its Supporting Information files. Complete results of the regression analyses are shown in S1 File.

**Funding:** This study was financially supported by Kirin Holdings Company, Limited (https://www.kirinholdings.com) in the form of a salaries for DN, KM, and MM, and in the form of awards received by TY, SI, and MK. No additional external funding was received for this study. The funder had no role in study design, data collection and analysis, decision to publish, or preparation of the manuscript.

**Competing interests:** The authors have read the journal's policy and have the following competing interests: DN, KM, and MM are employees of Kirin Holdings Company, Limited. This does not alter our adherence to PLOS ONE policies on sharing data and materials. There are no patents, products in development or marketed products associated with this research to declare.

NO production decreases vascular extensibility and causes various cardiovascular diseases such as myocardial infarction and cerebral apoplexy [1,2]. The NO cycle is one of the most important mechanisms for maintaining cardiovascular homeostasis. Moreover, decreased vascular function not only affects the cardiovascular system but also exerts a negative impact on the overall health of the body, including kidney, liver, and motor functions [3–5]. Although early improvement of age-related vascular dysfunction could contribute to cardiovascular health, vascular aging has few specific subjective symptoms until its functions worsen significantly, which prevents early intervention.

L-citrulline (Cit) is a nonessential amino acid found abundantly in watermelon (*Citrullus vulgaris*). Cit increases the production of NO that induces blood vessel dilation. NO is produced through a reaction catalyzed by endothelial NO synthase (eNOS) where L-arginine (Arg) is converted into Cit. Arg is regenerated from Cit via the Cit-NO cycle, and thus Cit effectively increases NO production. NO induces blood vessel dilation via the soluble guanylate cyclase (sGC)–cyclic guanosine monophosphate (cGMP) cascade [6]. Improving NO signaling activates cGMP-dependent protein kinase (PKG), thereby promoting the uptake of $Ca^{2+}$ into the sarcoplasmic reticulum, inducing the relaxation of vascular smooth muscle and decreasing intracellular $Ca^{2+}$ concentrations, and resulting in vasodilation.

Due to the ability of Cit to effectively produce NO, it is used as a nutritional supplement for vascular health and athletic performance [7]. Cit intake improves flow-mediated dilation, an indicator of vascular endothelial function [8], and brachial-ankle pulse wave velocity, an indicator of vascular stiffness [9]. However, the effects of Cit supplementation show individual differences, implying the existence of genetic variations in genes involved in the Cit metabolic pathway. Investigating such genetic variations is beneficial for identifying individuals for whom Cit is effective.

Single nucleotide polymorphism (SNP) is a major type of genetic variation that causes individual differences in a variety of traits [10]. A genetic association study is a powerful tool to detect the susceptibility of SNPs for a particular trait or disease [11]. SNPs identified by genetic association studies have a wide range of applications. For instance, in the medical and pharmaceutical industries, SNPs are used for predicting clinical risk and developing drugs [12]. They are also used in the food industry in response to the increasing focus on personalized nutrition [13]. An example is genotype-based nutritional supplementation, such as folic acid supplementation based on the *MTHFR* rs1801133 genotype [14] and n-3 PUFA supplementation based on the *FADS1* rs174546 genotype [15]. Regarding Cit and NO, the *NOS3* rs891511 genotype is associated with hypertension [16], and the *PDE3A* rs11045239 genotype is associated with endothelial function [17]. Nonetheless, there have been no studies on the associations between the SNPs of genes on the Cit metabolic and action pathways and the subjective symptoms of vascular aging. Such studies could result in the development of genotype-based supplementation of Cit, which would contribute to an efficient improvement of vascular aging. Furthermore, research on the traits and genetic factors associated with vascular aging could result in the development of markers to detect vascular aging at an early stage.

In this study, we evaluated the associations between SNPs on the Cit metabolic pathway and health characteristics related to vascular aging using a candidate gene approach [18]. We also conducted genome-wide association studies (GWAS) [19] to analyze SNPs more extensively across genes associated with Cit and NO metabolisms. The observed associations could be explained by the change of downstream signaling of NO, and the traits in the observed associations can be manifested as subjective symptoms of vascular aging. Therefore, the SNPs identified in this study could predict susceptibility to the subjective symptoms of vascular aging.

## Materials and methods

### Study participants and genotyping

We recruited 2000 Japanese men and women through email invitations sent to customers of MYCODE (DeNA Life Science, Inc., Tokyo, Japan), a personal genome service in Japan, between January 13 and February 28, 2023. Eligibility criterion was age > 40 years, and participation was voluntary. Saliva samples were collected from the customers and genotyped using the Infinium OmniExpress-24+ BeadChip or Human OmniExpress-24+ BeadChip (Illumina Inc., San Diego, CA, United States) as part of the MYCODE service. This study adhered to the principles outlined in the Declaration of Helsinki and was approved by the ethics committee of DeNA Life Science Inc. (protocol #20220722_1). It was registered with the University Hospital Medical Information Network in Japan (UMIN000048828). Written informed consent was obtained for MYCODE Research, a research platform based on the customers of MYCODE, in which the customers agreed for the use of their anonymized genetic data and/or health-related information for scientific research purposes. Additional informed consent for this specific study was obtained from all participants on the MYCODE website.

### Quality control and genotype imputation

Sample quality control (QC) steps were performed using PLINK version 1.9 [20]. Four samples identified as outliers were excluded from the Japanese cluster based on visual inspection of the top two principal components from genetic principal component analysis (PCA). No samples had sex inconsistency, missing call rate >1.0%, or identity by descent (PI_HAT) >0.1875. The final dataset included 1996 participants.

We conducted SNP QC on 694434 autosomal variants. SNPs with missing call rate >1.0%, Hardy–Weinberg equilibrium (HWE; $P < $1E-06), or minor allele frequency (MAF) <0.05 were excluded, after which 442537 autosomal variants remained. Genotype imputation was performed using Eagle version 2.4.1 [21] for the phasing step and Minimac3 version 2.0.1 [22] for imputation with East Asian samples (n = 504) from the 1000 Genomes Project Phase 3 [23] (1 KGP, n = 2504) imputation reference panel. Variants with low imputation quality (Rsq < 0.7) were excluded. In each association analysis, SNPs with missing call rate >1.0%, HWE ($P < $1E-06), or minor allele frequency (MAF) <0.05 were removed.

### Trait definition

Traits for association analyses were collected from the participants through an online questionnaire designed based on the relationship with vascular function. We used the POMS2 [24] and SF-36v2 [25] questionnaires, which are widely used to evaluate mood states and health-related quality of life (QOL), respectively. POMS2 evaluated the subjects mood, anxiety, and depression and can assess mood states from the seven scales of Anger-Hostility, Confusion-Bewilderment, Depression-Dejection, Fatigue-Inertia, Tension-Anxiety, Vigor-Activity, and Friendliness and the Total Mood Disturbance (TMD) scores that represent overall negative mood states. The median score for any of the items was calculated as 50 in the population studied, and scores >60 or <39 were considered more severely symptomatic. SF-36v2 was also calculated in the same way, with a median score of 50 within the population studied. Studies have reported that the POMS2 and SF-36v2 questionnaires are associated with flow-mediated dilation [26] and brachial-ankle pulse wave velocity [27], which are markers of vascular endothelial function and arterial stiffness, respectively. To evaluate traits not captured by POMS

and SF-36v2, we also used our own 29 visual analog scale (VAS) [28] items. The VAS items were designed to explore the subjective symptoms associated with impaired vascular function [29–32] and were expected to improve with Cit intake. These VAS items consisted of two endpoints, with 0 and 10 representing "feel not at all" and "feel extremely strongly," respectively. The POSM2 and SF-36v2 subscales or VAS questions used to define traits are listed in S1 Table with their mean score and variance. For each trait, subjects with worse scores than the threshold were defined as cases, and those with better scores were treated as controls, as described subsequently and summarized in S2 Table.

For POMS2, participants with a score of ≥51 who could be considered symptomatic were defined as cases, and those with ≤50 who could be considered not symptomatic were defined as controls for each subscale other than Vigor-Activity and Friendliness. For the Vigor-Activity and Friendliness subscales, participants with a score of ≤49 who could be considered symptomatic were defined as cases, and those with ≥50 who could be considered not symptomatic were defined as controls. For SF-36v2, participants with a score of ≤49 who could be considered symptomatic were defined as cases, and those with ≥50 who could be considered not symptomatic were defined as controls for each subscale. For all eight SF-36v2 subscales, we also defined a trait indicating a worse condition of not only health-related QOL but also a negative mood state measured using POMS2 TMD to investigate a more considerable status. For these traits, participants were first divided into two groups based on their POMS TMD scores, viz., a more symptomatic group with scores of ≥60 and a less symptomatic group with scores of ≤39. The mean SF-36 scores were calculated for each group and used as the threshold to define cases and controls for each SF-36 subscale. Specifically, the mean SF-36 score in the higher POMS TMD score group (≥60) was set as the threshold for cases, whereas that in the lower POMS TMD score group (≤39) was set for controls. Participants with SF-36 scores at or below the threshold for cases were defined as cases, whereas those with SF-36 scores at or above the threshold for controls were defined as controls. For VAS, cases and controls were defined by setting upper and lower thresholds for each question. Thresholds were set to include the top and bottom ~40 percentiles of the score distribution to separate the more and less symptomatic populations. Fifty-three traits were included in the association analyses (S2 Table).

## Selection of candidate genes and SNPs

We selected 20 candidate genes from the Cit-NO cycle and related pathways. From the Cit-NO cycle, we selected arginases *ARG1* (ENSG00000118520) and *ARG2* (ENSG00000081181), argininosuccinate lyase *ASL* (ENSG00000126522), argininosuccinate synthase *ASS1* (ENSG00000130707), dimethylarginine dimethylaminohydrolase *DDAH1* (ENSG00000153904), and nitric oxide synthase *NOS1* (ENSG00000089250). From the Cit synthesis pathway from glutamine, an upstream pathway of the Cit-NO cycle, we selected carbamoyl-phosphate synthase *CPS1* (ENSG00000021826), glutaminase *GLS* (ENSG00000115419), glutamate dehydrogenase *GLUD1* (ENSG00000148672), glutamate-ammonia ligase *GLUL* (ENSG00000135821), and N-acyl-L-amino-acid amidohydrolase *ABHD14A-ACY1* (ENSG00000248487). From the NO-cGMP cascade, a downstream pathway of the Cit-NO cycle, we selected sGC subunit *GUCY1A2* (ENSG00000152402); phosphodiesterases *PDE1A* (ENSG00000115252), *PDE2A* (ENSG00000186642), *PDE3A* (ENSG00000172572), *PDE5A* (ENSG00000138735), *PDE9A* (ENSG00000160191), and *PDE10A* (ENSG00000112541); and potassium calcium-activated channel *KCNMB4* (ENSG00000135643). We also selected adenylate cyclase *ADCY8* (ENSG00000155897), a major component of the cAMP pathway, because the cAMP pathway also affects calcium influx. Among the SNPs on the microarray, 597 SNPs with MAF ≥ 0.05 located on the candidate genes and its upstream region were used in association analyses. The selected SNPs and genes are listed in S3 Table.

## Association analyses

Logistic regression was performed on each trait using PLINK version 1.9. We applied an additive model of the minor allele and used age, sex, and PC1 and PC2 from the PCA of SNP data as covariates. For the candidate gene approach, we set a significance level at $P < 8.38E-05$ and a suggestive significance level at $P < 1.68E-04$ based on Bonferroni correction

[33]. For GWAS, we set a significance level at the genome-wide significance level (*P*<5E-08). The number of GWAS that analyzed SNPs for each trait is shown in S2 Table. The identified SNPs with the significance levels were annotated using SnpEff [34] based on GRCh37.75 [35] and investigated for cis-expression quantitative trait loci (eQTLs) effects using the Genotype-Tissue Expression (GTEx) Analysis Release V8 database [36].

## Results

### Study participants

We recruited 2000 Japanese men and women aged ≥40 years in this study. After QC, 1996 participants were included, among which there were 967 (48.4%) men and 1026 (51.4%) women, whereas 3 (0.2%) individuals did not report their sex (Table 1). The mean age of the participants was 53.5 years, and the age group 50–59 years was the most represented.

### Candidate gene approach

The associations between SNPs on the Cit metabolic pathway and traits that were selected based on their relationship with vascular function were evaluated using a candidate gene approach. The associations between the 597 SNPs and 53 traits were evaluated by logistic regression under the assumption of additive effects of the minor alleles by adjusting for age, sex, and PC1 and PC2 from the PCA of SNP data. Complete results of the regression analyses are shown in S1 File. After Bonferroni correction, four associations reached a significance level (Table 2), and four reached a suggestive significance level (Table 3). The associations that reached a significance level included those between *KCNMB4* rs17108108 C allele and tendency to gain weight, *ADCY8* rs6470860 G allele and numbness of limbs, *NOS1* rs2271987 T allele and lower back pain, and *PDE9A* rs2284972 G allele and body pain with negative mood states. Only *ADCY8* rs6470860 allele exhibited a protective association of the minor G allele with the trait, whereas the others exhibited associations in which their minor alleles increased the risk of traits (Fig 1). The associations that reached a suggestive significance level included those between *PDE10A* rs9365900 A allele and anger, *PDE1A* rs833141 T allele and warm body, *ADCY8*

**Table 1. Age and sex distribution of the participants.**

| Age group (years) | Male | % | Female | % | Unknown | % | Total |
|---|---|---|---|---|---|---|---|
| 40–49 | 317 | 44.6 | 393 | 55.3 | 1 | 0.1 | 711 |
| 50–59 | 352 | 44.0 | 446 | 55.8 | 2 | 0.3 | 800 |
| 60–69 | 245 | 59.0 | 170 | 41.0 | 0 | 0.0 | 415 |
| ≥70 | 53 | 75.7 | 17 | 24.3 | 0 | 0.0 | 70 |
| Total | 967 | 48.4 | 1026 | 51.4 | 3 | 0.2 | 1996 |

**Table 2. Significant associations in the candidate gene approach.**

| Trait | SNP | Effect allele | Position | Variant type | Gene | MAF | *P* value | OR | SE |
|---|---|---|---|---|---|---|---|---|---|
| Body pain with negative mood states | rs2284972 | G | chr21:44185993 | downstream, intronic | *PDE9A* | 0.420 | 6.39E-05 | 1.338 | 0.073 |
| Numbness of limbs | rs6470860 | G | chr8:131905190 | intronic | *ADCY8* | 0.290 | 8.10E-05 | 0.623 | 0.120 |
| | | | | upstream | *RP11-737F9.2* | | | | |
| Lower back pain | rs2271987 | T | chr12:117728517 | intronic | *NOS1* | 0.157 | 8.34E-05 | 1.499 | 0.103 |
| Tendency to gain weight | rs17108108 | C | chr12:70797917 | intronic | *KCNMB4* | 0.177 | 3.89E-05 | 1.613 | 0.116 |

MAF: minor allele frequency, *P* value: *P* value from the logistic regression analysis, OR: odds ratio, SE: standard error of OR.

**Table 3. Suggestive significant associations in the candidate gene approach.**

| Trait | SNP | Effect allele | Position | Variant type | Gene | MAF | P value | OR | SE |
|-------|-----|---------------|----------|--------------|------|-----|---------|-----|-----|
| Anger | rs9365900 | A | chr6:166064152 | intronic | *PDE10A* | 0.283 | 1.34.E-04 | 0.732 | 0.082 |
| Warm body | rs833141 | T | chr2:183162809 | intronic | *PDE1A* | 0.071 | 1.10.E-04 | 0.514 | 0.172 |
| Numbness of limbs | rs2164306 | A | chr8:131887274 | intronic | *ADCY8* | 0.336 | 1.53.E-04 | 0.653 | 0.113 |
| Tendency to think about this and that when asleep | rs6665825 | A | chr1:85955261 | intronic | *DDAH1* | 0.103 | 1.31.E-04 | 0.558 | 0.153 |

MAF: minor allele frequency, P value: P value from the logistic regression analysis, OR: odds ratio, SE: standard error of OR.

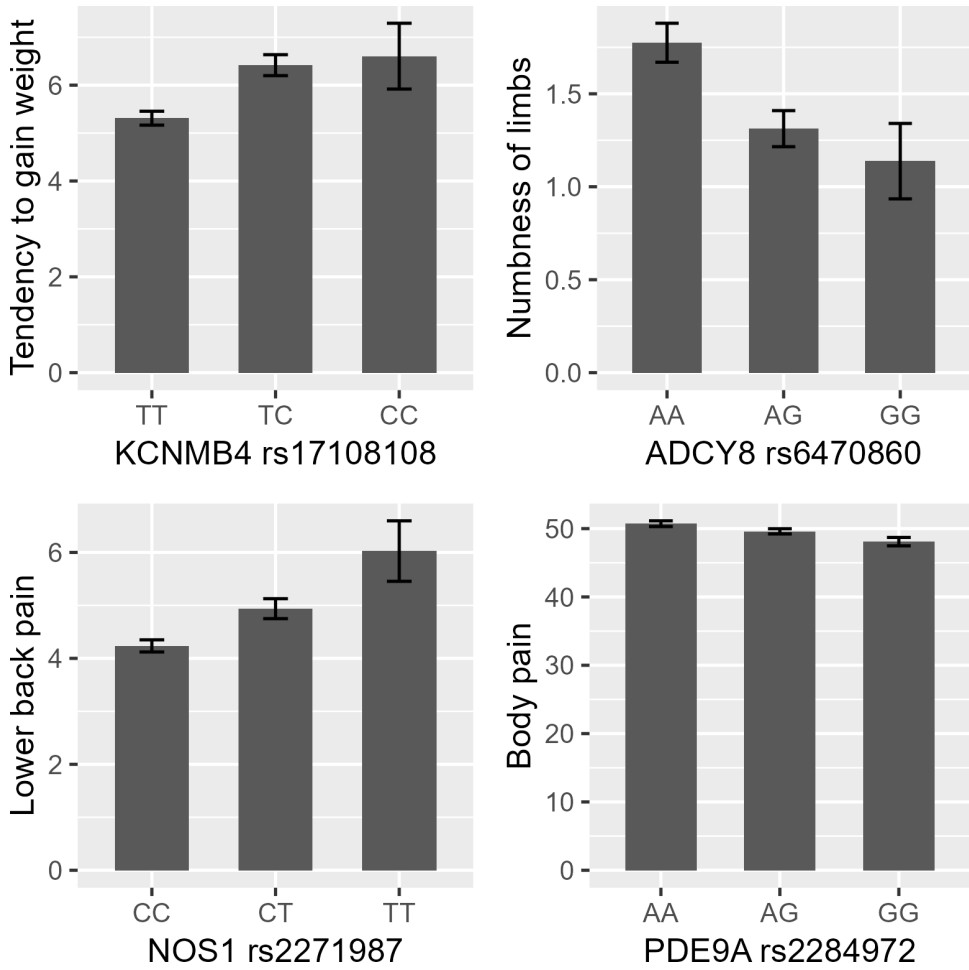

**Fig 1. Comparison of questionnaire scores among genotypes for the significant associations.** To facilitate an intuitive understanding of the effect of SNPs in the candidate gene analysis, the means for each genotype were visualized as a bar graph. The mean questionnaire scores were compared across the SNP genotypes for those with significant associations with the corresponding traits. For both traits, the population with homozygous minor allele was the most symptomatic, followed by the heterozygous (major and minor alleles) population. The x-axis represents the genotypes for each SNP, arranged from left to right as follows: homozygous major allele, heterozygous, and homozygous minor allele. The y-axis shows either the mean of SF-36 scores or VAS scores used for defining each trait. For body pain, SF-36 scores are shown, where lower scores indicate more severe pain. For all other traits, VAS scores are displayed. Error bars indicate standard error of mean.

rs2164306 A allele and numbness of limbs, and *DDAH1* rs6665825 A allele and tendency to think about this and that when asleep.

## GWAS

We also conducted GWAS of the 53 traits to analyze SNPs more extensively across genes associated with Cit and NO metabolisms. Manhattan plots and quantile–quantile plots for all the 53 traits are shown in S1 Fig. A total of 13 SNPs spanning the three genomic regions demonstrated associations at the genome-wide significance level (Table 4). The minor C allele of rs12028323, located at the *PRMT6* (ENSG00000198890) and *NTNG1* (ENSG00000162631) intergenic region, was associated with mood disturbance. The minor allele G of rs11106113 was associated with poor physical func-tioning, and rs10022096 and its proxy SNPs (the SNPs in moderate-to-high LD ($R^2 > 0.6$) with the lead SNP), which were located near *HSD17B11* (ENSG00000198189), were associated with role limitations due to emotional problems. Among the mapped genes, only *PRMT6* was associated with vascular function (see Discussion). We then focused on *PRMT6-NTNG1* rs12028323 for subsequent eQTL analysis.

## eQTL analysis

To gain insights into the molecular mechanisms underlying the observed associations, we searched eQTLs in the GTEx database for SNPs that demonstrated significance or suggestive significance levels in the association analyses. Among the eight SNPs identified in the candidate gene approach, rs17108108 C allele and rs6470860 G allele were identified as eQTLs associated with increased expression level of *KCNMB4* in various tissues (fibroblasts cells showed the lowest *P* value) and decreased expression level of *ADCY8* in the testis, respectively (Table 5). For GWAS-identified SNPs, we searched eQTLs for *PRMT6-NTNG1* rs12028323, but no eQTLs were found. Then, we also searched eQTLs for proxy SNPs, the SNPs in moderate-to-high LD ($R^2 > 0.6$) with rs12028323, with the suggestive level ($P < 1E-05$) of associations in GWAS. We detected 44 SNPs as eQTLs. The GTEx database contained 924 entries for these 44 SNPs, of which 902

**Table 4. Significant associations in GWAS.**

| Trait | SNP | Effect allele | Position | Variant type | Gene | MAF | *P* value | OR | SE |
|---|---|---|---|---|---|---|---|---|---|
| Mood disturbance | rs12028323 | C | chr1:107627510 | intergenic | *PRMT6-NTNG1* | 0.096 | 2.43E-08 | 1.891 | 0.114 |
| Poor physical functioning | rs11106113 | G | chr12:91758350 | intronic | *RP11-121E16.1* | 0.455 | 4.51E-08 | 1.547 | 0.080 |
| Poor physical functioning with negative mood states | rs11106113 | G | chr12:91758350 | intronic | *RP11-121E16.1* | 0.455 | 4.71E-08 | 1.692 | 0.096 |
| Role limitations due to emo-tional problems | rs10022096 | C | chr4:88279174 | intronic | *HSD17B11* | 0.422 | 8.83E-09 | 1.850 | 0.107 |
| | rs6531979 | A | chr4:88280101 | intronic | *HSD17B11* | 0.422 | 8.83E-09 | 1.850 | 0.107 |
| | rs6845304 | T | chr4:88280502 | intronic | *HSD17B11* | 0.431 | 2.92E-08 | 1.811 | 0.107 |
| | rs4438729 | G | chr4:88280826 | intronic | *HSD17B11* | 0.424 | 1.65E-08 | 1.827 | 0.107 |
| | rs9993968 | G | chr4:88281099 | intronic | *HSD17B11* | 0.424 | 1.65E-08 | 1.827 | 0.107 |
| | rs6853049 | C | chr4:88282021 | intronic | *HSD17B11* | 0.424 | 1.65E-08 | 1.827 | 0.107 |
| | rs4443241 | C | chr4:88288599 | upstream, intronic | *HSD17B11* | 0.431 | 9.81E-09 | 1.851 | 0.107 |
| | rs10002620 | G | chr4:88289252 | upstream, intronic | *HSD17B11* | 0.431 | 9.81E-09 | 1.851 | 0.107 |
| | rs6531983 | G | chr4:88291233 | upstream, down-stream, intronic | *HSD17B11* | 0.431 | 9.81E-09 | 1.851 | 0.107 |
| | rs6531984 | C | chr4:88291535 | upstream, down-stream, intronic | *HSD17B11* | 0.431 | 9.81E-09 | 1.851 | 0.107 |

MAF: minor allele frequency, *P* value: *P* value from the logistic regression analysis, OR: odds ratio, SE: standard error of OR.

**Table 5. eQTL analysis of significant SNPs.**

| SNP | Effect allele | Gene | P value | NES | SE | Tissue |
|---|---|---|---|---|---|---|
| rs17108108 | C | KCNMB4 | 2.44E-19 | 0.44 | 0.05 | Cells - Cultured fibroblasts |
| rs17108108 | C | KCNMB4 | 5.78E-11 | 0.43 | 0.06 | Whole Blood |
| rs17108108 | C | KCNMB4 | 8.43E-08 | 0.34 | 0.06 | Skin - Sun Exposed (Lower leg) |
| rs17108108 | C | KCNMB4 | 1.51E-05 | 0.31 | 0.07 | Skin - Not Sun Exposed (Suprapubic) |
| rs6470860 | A | ADCY8 | 9.28E-06 | -0.20 | 0.04 | Testis |

P value: P value from single-tissue eQTLs of the GTEx database, NES: normalized effect size, SE: standard error of NES.

demonstrated associations with increased expression levels of *PRMT6* and *NTNG1* in various tissues (S4 Table). Among them, rs9662586, rs11185057, and rs12021578 were located on the upstream regions of *PRMT6*.

## Discussion

We evaluated the association of SNPs on the Cit metabolic and action pathway with health characteristics related to vascular aging. We detected four significant associations and four suggestive significant associations in the candidate gene approach (Tables 2 and 3) and 13 significant associations in GWAS (Table 4). In the candidate gene approach, the associations that reached a significance level included those between the *KCNMB4* rs17108108 C allele and tendency to gain weight, the *ADCY8* rs6470860 G allele and numbness of limbs, the *NOS1* rs2271987 T allele and lower back pain, and *PDE9A* rs2284972 G allele and body pain with negative mood states. The key associations observed in our study are summarized in Fig 2.

*KCNMB4* encodes potassium calcium-activated channel subfamily M regulatory beta subunit 4. *KCNMB4* activates BK channel, a calcium-dependent potassium channel, that results in decreased intracellular $Ca^{2+}$ levels through suppression of $Ca^{2+}$ channel [37,38]. As depletion of $Ca^{2+}$ in fat cells causes abnormality in insulin secretion, we hypothesized that high expression levels of *KCNMB4* in fat cells cause altered insulin secretion through the activation of BK channel and subsequent decreased $Ca^{2+}$ levels [39]. The rs17108108 C allele was identified as an eQTL associated with increased expression level of *KCNMB4* in fibroblasts cells, precursors of fat cells (Table 5). Therefore, rs17108108 C allele carrier may be susceptible to altered insulin secretion that is caused by decreased levels of $Ca^{2+}$ in fat cells. As abnormality in insulin secretion induces obesity [40], the tendency to gain weight in rs17108108 C allele carriers may be partially explained by altered insulin secretion caused by decreased levels of $Ca^{2+}$. Furthermore, increased $Ca^{2+}$ levels in the vascular tissue cause vasoconstriction that in turn causes obesity. Although the eQTL effect of rs17108108 in the vascular tissue is unclear, assuming that the rs17108108 C allele decreases the expression of *KCNMB4* in the vascular tissue, this allele may induce the tendency to gain weight via vasoconstriction caused by increased $Ca^{2+}$ levels.

*ADCY8* is an ADCY that catalyzes the conversion of adenosine triphosphate (ATP) into 3′,5′-cyclic adenosine monophosphate (cAMP). cAMP induces vascular dilation through a different pathway than the NO-cGMP cascade. cAMP activates PKA, which decreases $Ca^{2+}$ levels in endothelial cells, resulting in vascular dilation [41–43]. Based on this, we hypothesized that low *ADCY8* expression levels result in increased $Ca^{2+}$ levels in endothelial cells, causing susceptibility to vasoconstriction. Reduced blood flow due to decreased vascular function causes numbness and pain. As vasoconstriction causes temporary ischemia that could result in numbness, low *ADCY8* expression levels and subsequent increased $Ca^{2+}$ levels in endothelial cells may increase susceptibility to numbness. The rs6470860 A allele (alternative allele) was identified as an eQTL associated with decreased expression level of *ADCY8* in the testis (Table 5). The rs6470860 G allele (reference allele) was negatively associated with numbness of limbs, implying that A allele carriers felt more numbness. Although the effect of rs6470860 on *ADCY8* expression in endothelial cells remains unclear, the numbness in

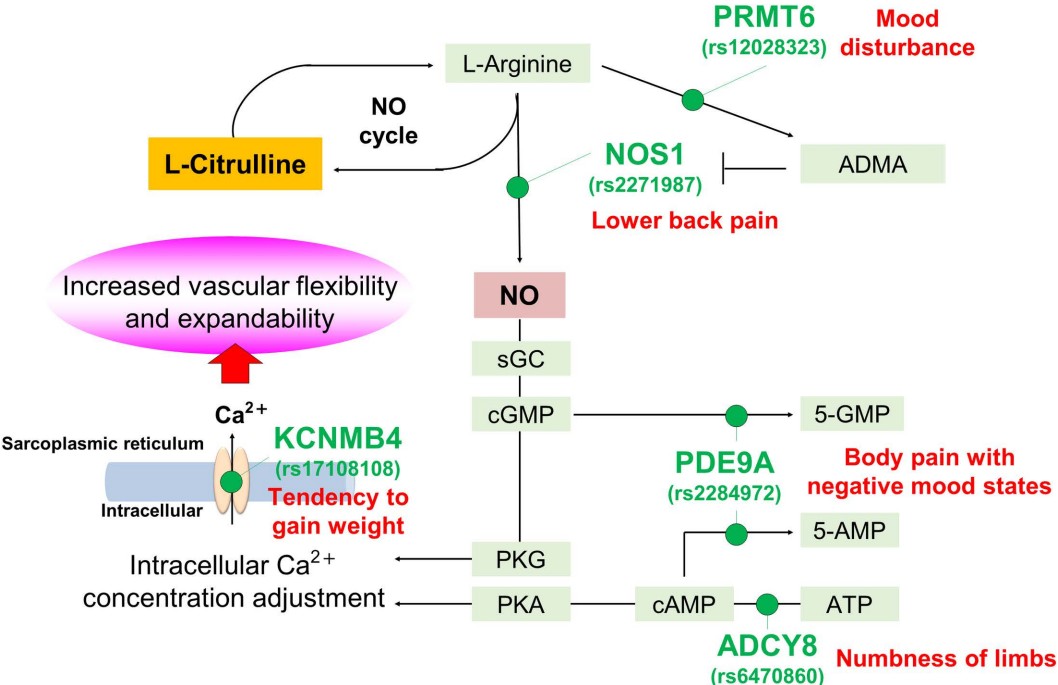

**Fig 2. Summary of sites of activity of genes encoded by SNPs observed to be associated in our study in the NO signaling pathway downstream of Cit.** The sites of activity of genes containing SNPs observed to be associated in our study in the pathway where Cit contributes to vascular flexibility and expandability (NO signaling pathway) are indicated. Genes with observed associations (SNPs that were associated in parentheses) are indicated in green letters, and symptoms of vascular aging that were observed to be associated with that gene (SNP) are indicated near the gene name in red letters. Arg conversion to NO is the first step in the NO signaling pathway, and it is critical to increase the activity of NOS and inhibit the activity of protein Arg methyltransferase inhibitor (PRMT) in increasing NO production. In addition, downstream NO signaling must lead to sGC-cGMP-mediated regulation of intracellular Ca2 + concentrations via PKG or protein kinase A (PKA) to correctly contribute to vascular flexibility and expandability. PDE, ADCY, and KCNMB play an important role in these pathways. NOS: NO synthase, sGC-cGMP: soluble guanylate cyclase –cyclic guanosine monophosphate, PKG: cGMP-dependent protein kinase, PDE: Phosphodiesterase, ADCY: Adenylyl cyclases.

rs6470860 A allele carrier may be partially explained by the increased $Ca^{2+}$ level in endothelial cells caused by low *ADCY8* expression levels.

*NOS1* is a neuronal nitric oxide synthase (nNOS) that catalyzes the conversion of Arg into NO. nNOS has been implicated in neurodegenerative diseases and the neuroregulation of smooth muscle, including sphincter relaxation [44]. In skeletal muscle cells, NO produced by nNOS increases $Ca^{2+}$ levels via TRPV1 activation, which activates mTOR and thus causes muscle hypertrophy [45]. Based on this, we hypothesized that low *NOS1* levels result in relatively weak muscle strength and cause lower back pain. Although no eQTLs were detected for rs2271987 in the GTEx database, which primarily consists of data from European populations, assuming that the rs2271987 T allele is associated with decreased *NOS1* expression levels in the skeletal muscle in the Japanese population, the lower back pain in rs2271987 T allele carriers may be partially explained by the relatively weak muscle strength caused by low *NOS1* expression levels.

*PDE9A* is a PDE subtype that regulates cGMP and cAMP signaling. PDE is primarily involved in the degradation of cGMP or cAMP. cGMP and cAMP signaling is essential in regulating the diameter of the cerebral artery, and the activation of *PDE9A* is associated with neurovascular migraine and ischemic stroke [46,47]. Migraine is a headache syndrome of unknown etiology; however, activation of the trigeminovascular pain signaling system appears to be involved in most types of migraine [48,49]. This pain signaling ascends from the cerebral blood vessels in the meninges, including the venous sinuses, basilar artery, and middle cerebral artery, perhaps activated by vasodilation or plasma extravasation. The pain-mediating afferent fibers in the trigeminal nerve transmit pain signal to the nerve cell bodies of the trigeminal

ganglion [50]. From the trigeminal nerve ganglion, the signal is further propagated to the spinal trigeminal nucleus in the brain stem, contralateral thalamic nuclei, and finally somatosensory cortex, where pain is perceived [51]. Although no eQTLs were detected for rs2284972, assuming that the rs2284972 G allele is associated with increased *PDE9A* expression levels in the brain and vascular tissues, body pain with negative mood states in rs2284972 G allele carriers may be partially explained by the constriction of cerebral vascular diameter and chronic dysfunction of cranial nerves caused by high *PDE9A* expression levels.

Although candidate gene analysis has higher power and analytical sensitivity and allows the detailed evaluation of SNPs in specific genes based on previous studies, it could miss other important SNPs. Therefore, we also conducted GWAS to analyze SNPs more extensively across genes related to Cit and NO metabolisms. Associations between the rs12028323 C allele and mood disturbance reached genome-wide significance level (Table 4). rs12028323 is located at the intergenic region of *PRMT6* and *NTNG1*. To our knowledge, no relationship between *NTNG1* and vascular function or Cit has been reported. *PRMT6* is a protein Arg methyltransferase that catalyzes monomethylation or dimethylation of Arg residues. Degradation of proteins with methylated Arg residues results in the production of free asymmetric dimethylarginine (ADMA). Because ADMA acts as a strong inhibitor of NOS and prevents NO production, increased *PRMT6* expression levels may result in susceptibility to vasoconstriction by increasing ADMA production and subsequent decreased NO production [52,53]. No eQTLs were detected for rs12028323 in the GTEx database, whereas three proximal SNPs, rs11185057, rs12021578, and rs9662586, were identified as eQTLs for *PRMT6*. These three SNPs were in moderate LD with rs12028323 (D′ = 1 and $R^2$ = 0.6351 in 1 KGP Japanese population), showed the suggestive significance level (P < 1E-05) of associations in GWAS, and were located on the upstream regions of PRMT6 (S4 Table). The effect alleles of the three SNPs, which corresponds to the rs12028323 C allele, associated with increased expression levels of *PRMT6* in various tissues. These results are consistent with our hypothesis that the eQTL-mediated upregulation of *PRMT6* results in decreased NO production. A possible reason why the top associated SNP, rs12028323, was not identified as an eQTL is its quite low frequency in European populations (C = 0.0003 in the ALFA project), which were largely used in the GTEx project. Based on these findings, we assumed that the rs12028323 C allele also exerts a positive effect on the expression level of *PRMT6*. As reported by a previous study that an increased TMD score is associated with decreased vascular endothelial function [26], mood disturbance in rs12028323 C allele carriers may be partially explained by vasoconstriction caused by higher expression levels of *PRMT6* and subsequent increased ADMA production. We also observed associations between rs11106113 and poor physical functioning, rs11106113 and poor physical functioning with negative mood states, and the 10 SNPs mapped on *HSD17B11* and role limitations due to emotional problems. A study investigating the gene expression profile in the heart by subcutaneously injecting rats with nitroglycerin, known to have vasodilating properties, reported a significant decrease in HSD17B11 expression [54]. This suggests that vasomodulatory action may still be involved behind the SNPs on HSD17B11 and symptoms that showed association in this study. However, to our knowledge, no relationship between Cit and NO has been reported for the nearest genes for these SNPs and the genes on which these SNPs exerted eQTL effects.

This study had several limitations. It lacks an independent replication. Further independent replication studies are required to validate our findings. Because GWAS results are not always replicated in other populations or races [55], future validation in different populations and races is needed to better ensure the association between the identified SNPs and the symptoms related to vascular aging toward generalizing applicability. For GWAS, we used a relatively small sample size (n = 555–1993) compared with current GWAS standards (tens or hundreds of thousands), which is another limitation. Future studies with larger sample sizes would be desirable to improve the statistical power and generalizability of our findings. We adjusted for well-established confounding factors, including age, sex, and PC1 and PC2. However, the possibility of residual confounding due to unmeasured or unknown factors cannot be ruled out. For the eQTL analysis, we used the GTEx database that primarily consists of data from European populations. The lack of a comprehensive Japanese eQTL database limits our ability to completely validate the functional relevance of the SNPs identified in our study.

The traits were derived from self-reported questionnaires, which provided an appropriate method for evaluating the subjective symptoms of vascular aging. Nevertheless, this approach may introduce misclassification, a recognized limitation in self-reported data acquisition. Moreover, although traits can be manifested as subjective symptoms of vascular aging, these subjective symptoms are not specific to vascular aging and are common to other unhealthy conditions, preventing people from realizing vascular aging only based on the symptoms.

Despite these limitations, most associations observed in this study could be explained by the change in downstream signaling of NO, supporting the association between the investigated traits and vascular aging. Future studies may possibly determine an individual's risk of vascular aging and Cit requirement by measuring SNPs found to be associated in this study, for example, by genetic testing, which might offer promising personalized supplements for older people. Cit improves NO production *in vivo* as well as upregulates sGC-cGMP signaling downstream of NO [56,57]. These signals are ultimately involved in regulating $Ca^{2+}$ levels in vascular endothelial cells, thereby controlling vascular contraction and relaxation. Our results allow us to propose the potential effects of Cit supplementation on the holders of traits and genotypes involved in the observed associations because Cit intake affects the downstream signaling of NO involved in regulating $Ca^{2+}$ levels and demonstrates effectiveness on vascular function [7–9]. To further develop this study, it could be a potential therapeutic approach in cardiovascular health to verify whether vascular aging occurs by collecting a population that possesses SNPs associated with the subjective symptoms of vascular aging in this study and evaluate the efficacy of Cit intake in these populations for future applications to personalized interventions for vascular aging. Overall, we propose that the identified SNPs predict susceptibility to the subjective symptoms of vascular aging, and the genotype-based supplementation of Cit contributes to an efficient improvement of vascular aging.

## Supporting information

**S1 Table. Summary of subscales or questions utilized to define traits.** SD: standard deviation of mean, SE: standard error of mean, L95: lower 95% confidence interval, U95: upper 95% confidence interval.
(XLSX)

**S2 Table. Definitions of traits for association analyses.**
(XLSX)

**S3 Table. 597 SNPs for the candidate gene approach.**
(XLSX)

**S4 Table. eQTL analysis for proxy SNPs of *PRMT6-NTNG1* rs12028323.** $R^2$: squared correlation coefficient with rs12028323, OR: odds ratio, SE: standard error, L95: lower 95% confidence interval, U95: upper 95% confidence interval, NES: normalized effect size.
(XLSX)

**S1 Fig. Manhattan plots and Quantile–Quantile (QQ) plots for GWAS.** Each trait is indicated in the top left corner of the respective plot. The left plot shows the Manhattan plot, whereas the right plot shows the QQ plot. The red line on the Manhattan plot represents the genome-wide significance level ($P < $5E-08), and the blue line represents the genome-wide suggestive significance level ($P < $1E-05).
(PDF)

**S1 File. Complete results of regression analyses of the candidate gene approach.** This file is a modified version of PLINK's output file (.assoc.logistic), where a new first column containing trait information has been inserted, and the file format has been converted into CSV. CHR: chromosome, BP: base-pair coordinate, A1: minor allele, TEST: PLINK test

identifier, NMISS: number of observations, OR: odds ratio, SE: standard error of OR, L95: lower 95% confidence interval, U95: upper 95% confidence interval, STAT: T-statistic, P: asymptotic P value for t-statistic.
(CSV)

## Acknowledgments

We gratefully thank the customers of MYCODE who participated in this study.

## Author contributions

**Conceptualization:** Dai Nogimura, Kazuki Moriyasu, Masahiko Morita.

**Data curation:** Sachiko Ishida, Masakazu Kohda, Takayuki Yazawa.

**Formal analysis:** Sachiko Ishida, Masakazu Kohda, Takayuki Yazawa.

**Project administration:** Masahiko Morita.

**Resources:** Dai Nogimura, Sachiko Ishida, Masakazu Kohda, Takayuki Yazawa.

**Supervision:** Masahiko Morita.

**Visualization:** Dai Nogimura.

**Writing – original draft:** Dai Nogimura.

**Writing – review & editing:** Dai Nogimura, Sachiko Ishida, Masakazu Kohda, Takayuki Yazawa, Masahiko Morita.

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
