## [Decision Letter · Decision Letter 0]

17 Jan 2025

PONE-D-24-52040SNPs on the L-citrulline metabolic pathway are associated with specific health characteristics related to vascular aging in the Japanese populationPLOS ONE

Dear Dr. Morita,

Thank you for submitting your manuscript to PLOS ONE. After careful consideration, we feel that it has merit but does not fully meet PLOS ONE’s publication criteria as it currently stands. Therefore, we invite you to submit a revised version of the manuscript that addresses the points raised during the review process.

We look forward to receiving your revised manuscript.

Kind regards,

Asif Jan, Ph.D

Academic Editor

PLOS ONE

**Journal Requirements:**

Reviewers' comments:

Reviewer's Responses to Questions

**Comments to the Author**

1. Is the manuscript technically sound, and do the data support the conclusions?

Reviewer #1: Yes

Reviewer #2: Yes

2. Has the statistical analysis been performed appropriately and rigorously? 

Reviewer #1: Yes

Reviewer #2: Yes

3. Have the authors made all data underlying the findings in their manuscript fully available?

Reviewer #1: Yes

Reviewer #2: Yes

4. Is the manuscript presented in an intelligible fashion and written in standard English?

Reviewer #1: Yes

Reviewer #2: Yes

5. Review Comments to the Author

**Reviewer #1:**  Minor comments:-The manuscript titled "SNPs on the L-citrulline Metabolic Pathway Are Associated with Specific Health Characteristics Related to Vascular Aging in the Japanese Population" presents an intriguing study on the genetic factors influencing vascular aging. The focus on nitric oxide production and L-citrulline's role in improving vascular function is highly relevant to cardiovascular research. The integration of both candidate gene and genome-wide association approaches strengthens the findings.

However, the manuscript can benefit from clearer articulation of the clinical implications of these findings. The discussion should expand on how these genetic associations can be translated into personalized interventions for vascular aging. Additionally, the limitations section is comprehensive, but it would be helpful to discuss potential future directions, such as validating these findings in other populations.

Overall, this is a well-conducted study with significant potential, but further refinement in linking the genetic associations to practical applications will enhance the manuscript’s impact.

**Reviewer #2: ** 1. Clinical Relevance and Practical Implications

The manuscript provides significant findings on SNPs associated with vascular aging,

but the discussion should emphasize how these genetic markers can be translated into

clinical practice. For instance, the authors could elaborate on the potential for

genotype-based supplementation of L-citrulline to improve vascular health in aging

populations. Including more practical applications would make the study more

impactful.

2. Broader Applicability of Findings

The study focuses exclusively on the Japanese population. It would be beneficial to

discuss the generalizability of these findings to other ethnic groups. The authors could

also mention the importance of validating these results in diverse populations to

ensure broader applicability.

3. GWAS Results Interpretation

The manuscript includes genome-wide association study (GWAS) findings, but the

interpretation of the identified SNPs could be more detailed. The authors should

clarify the functional relevance of these SNPs and how they impact downstream nitric

oxide signaling, particularly for the newly identified associations.

4. Methodological Details

While the methodology is described in detail, some sections could benefit from

additional clarity. For example, the selection criteria for the traits used in association

analyses should be more explicitly explained. Additionally, the rationale behind using

both candidate gene and GWAS approaches should be stated more clearly.

Title

The title is informative but could be refined for clarity. Consider simplifying it to:

“SNP Associations in the L-Citrulline Pathway and Vascular Aging in the Japanese

Population.”

2. Abstract

The abstract is comprehensive but slightly dense. Simplifying some of the

terminology for a broader audience would improve readability. Additionally, the

abstract should briefly highlight the practical implications of the study findings.

3. Figures and Tables

The figures and tables are well-organized, but the legends could be more descriptive.

Providing additional context for each figure would help readers better understand the

results at a glance.

4. Terminology Consistency

Ensure consistency in using abbreviations and technical terms throughout the

manuscript. For example, "L-citrulline" should be consistently abbreviated as "Cit"

after its first use.

6. PLOS authors have the option to publish the peer review history of their article (what does this mean? ). If published, this will include your full peer review and any attached files.

**Do you want your identity to be public for this peer review?** For information about this choice, including consent withdrawal, please see our Privacy Policy .

Reviewer #1: No

Reviewer #2: **Yes: ** DR.WAHEED ALI SHAH

---

## [Author Response · Author response to Decision Letter 1]

17 Feb 2025

Dear Prof. Emily Chenette, Editor-in-Chief, PLoS ONE

Manuscript Number: PONE-D-24-52040

Title: SNPs on the L-citrulline metabolic pathway are associated with specific health characteristics related to vascular aging in the Japanese population

Authors: Dai Nogimura, Kazuki Moriyasu, Sachiko Ishida, Masakazu Kohda, Takayuki Yazawa, Masahiko Morita

We would like to thank the two reviewers for their valuable comments, which we have used to revise our manuscript accordingly. These comments have helped us significantly improve our paper. Please see our revised manuscript and point-by-point response to the reviewers’ comments. We are writing the modified parts in red, and these parts were checked by a native English speaker. The authors felt the revised manuscript is a suitable response to the comments and has significantly improved over the initial submission. Thank you very much for your consideration and kind assistance.

Sincerely yours,

Masahiko Morita, Ph. D.

Institute of Health Sciences, Kirin Holdings Company, Ltd.

2-26-1, Muraoka-Higashi, Fujisawa, Kanagawa 251-8555, Japan

Phone No: +81-90-3063-2127

Email Address: Masahiko_Morita@kirin.co.jp

Response to the reviewers’ comments.

Comment from Reviewer #1

The discussion should expand on how these genetic associations can be translated into personalized interventions for vascular aging.

AND,

Comment from Reviewer #2

1. Clinical Relevance and Practical Implications

The manuscript provides significant findings on SNPs associated with vascular aging, but the discussion should emphasize how these genetic markers can be translated into clinical practice. For instance, the authors could elaborate on the potential for genotype-based supplementation of L-citrulline to improve vascular health in aging populations. Including more practical applications would make the study more impactful.

Response:

We strongly agree with the reviewer that the practical implications of the genetic associations in this study need to be considered. As you indicated, the results of this study support the validity of genotype-based Cit supplementation in the future, which might make personalized supplements beneficial. We have added a statement including the possibility of further research development in the final paragraph of the Discussion section (page 20-21, lines 472–476, 480, and 482–486).

Comment from Reviewer #1

The limitation section is comprehensive, but it would be helpful to discuss potential future directions, such as validating these findings in other populations.

AND,

Comment from Reviewer #2

2. Broader Applicability of the Findings

The study focuses exclusively on the Japanese population. It would be beneficial to discuss the generalizability of these findings to other ethnic groups. The authors could also mention the importance of validating these results in diverse populations to ensure broader applicability.

Response:

Thank you for your comments. As you indicated, future validation in different populations and races is needed to better ensure the association between the SNPs found in this study and the symptoms of vascular aging. We have added this point in the seventh paragraph of the Discussion section (page 20, lines 449–453).

Comment from Reviewer #2

3. GWAS Results Interpretation

The manuscript includes the genome-wide association study (GWAS) findings, but the interpretation of the identified SNPs could be more detailed. The authors should clarify the functional relevance of these SNPs and the way they impact downstream nitric oxide signaling, particularly for the newly identified associations.

Response:

Thank you for pointing this out. The interpretation of the SNPs identified in the GWAS is provided in paragraph 6 of the Discussion section. We have already discussed PRMT6 in depth for its involvement with the NO signaling pathway. In response to your suggestion, we have also added the gene for the other identified SNP (HSD17B11) on page 19, lines 439 to 444, regarding its possible association with vascular aging. However, as mentioned on the same page in lines 444 to 446, no association with the Cit and NO action pathways has been reported.

Comment from Reviewer #2

4. Methodological Details

While the methodology is described in detail, some sections could benefit from additional clarity. For example, the selection criteria for the traits used in association analyses should be more explicitly explained. Additionally, the rationale behind using both the candidate gene and GWAS approaches should be stated more clearly.

Response:

Thank you for this suggestion. I will first reply to the first half of your point. We used POMS2 and SF-36v2 and our visual analog scale (VAS) to determine the trait of association analysis in this study. We have added the detailed reasons for selecting each investigation method and the case/control selection criteria in the Trait definition section on page 6-8 to make it clearer. We will then respond to the latter part of your comment. Although candidate gene analysis has high power and is useful for finding associations between SNPs in specific genes and symptoms, it may miss other important SNPs. Therefore, we also performed GWAS to more broadly analyze SNPs across genes related to Cit/NO metabolism and the trait. We have added the rationale for using the candidate gene and the GWAS approach in the sixth paragraph of the Discussion section (page 18, lines 408–410).

Comment from Reviewer #2

Title

The title is informative but could be refined for clarity. Consider simplifying it to “SNP Associations in the L-Citrulline Pathway and Vascular Aging in the Japanese Population.”

Response:

Thank you for pointing this out. We have revised the title based on your suggestion to make it simpler.

Comment from Reviewer #2

Abstract

The abstract is comprehensive but slightly dense. Simplifying some of the terminology for a broader audience would improve readability. Additionally, the abstract should briefly highlight the practical implications of the study findings.

Response:

Thank you for pointing this out. We have simplified the sentences and phrases to make the Abstract more readable and added a note of the practical implications of the study findings at the end.

Comment from Reviewer #2

Figures and Tables

The figures and tables are well-organized, but the legends could be more descriptive. Providing additional context for each figure would help readers better understand the results at a glance.

Response:

In response to your suggestion, we have added background explanations to the legend of each figure for better understanding by the reader. In Fig 1 legend, we have added a brief description of the main purpose of the figure and the results. In Fig 2 legend, we have added a description of the role of the genes encoded by SNPs observed to be associated with the Cit and NO pathways in our study.

Comment from Reviewer #2

Terminology Consistency

Ensure consistency in using abbreviations and technical terms throughout the manuscript. For example, "L-citrulline" should be consistently abbreviated as "Cit" after its first use.

Response:

Thank you very much for your direction. Again, we have checked the consistency of terminology throughout the manuscript and corrected the inadequacies.

---

## [Editor Report · Decision Letter 1]

10 Mar 2025

PONE-D-24-52040R1SNP associations in the L-citrulline metabolic pathway and vascular aging in the Japanese populationPLOS ONE

Dear Dr. Morita,

Thank you for submitting your manuscript to PLOS ONE. After careful consideration, we feel that it has merit but does not fully meet PLOS ONE’s publication criteria as it currently stands. Therefore, we invite you to submit a revised version of the manuscript that addresses the points raised during the review process.

We look forward to receiving your revised manuscript.

Kind regards,

Asif Jan, Ph.D

Academic Editor

PLOS ONE
---

## [Author Response · Author response to Decision Letter 2]

31 Mar 2025

In addition to our previous response, we have added a statement regarding updating reference information.

Dear Prof. Emily Chenette, Editor-in-Chief, PLoS ONE

Manuscript Number: PONE-D-24-52040

Title: SNPs on the L-citrulline metabolic pathway are associated with specific health characteristics related to vascular aging in the Japanese population

Authors: Dai Nogimura, Kazuki Moriyasu, Sachiko Ishida, Masakazu Kohda, Takayuki Yazawa, Masahiko Morita

We would like to thank the two reviewers for their valuable comments, which we have used to revise our manuscript accordingly. These comments have helped us significantly improve our paper. Please see our revised manuscript and point-by-point response to the reviewers’ comments. We are writing the modified parts in red, and these parts were checked by a native English speaker. The authors felt the revised manuscript is a suitable response to the comments and has significantly improved over the initial submission. Thank you very much for your consideration and kind assistance.

Sincerely yours,

Masahiko Morita, Ph. D.

Institute of Health Sciences, Kirin Holdings Company, Ltd.

2-26-1, Muraoka-Higashi, Fujisawa, Kanagawa 251-8555, Japan

Phone No: +81-90-3063-2127

Email Address: Masahiko_Morita@kirin.co.jp

Response to the reviewers’ comments.

Comment from Reviewer #1

The discussion should expand on how these genetic associations can be translated into personalized interventions for vascular aging.

AND,

Comment from Reviewer #2

1. Clinical Relevance and Practical Implications

The manuscript provides significant findings on SNPs associated with vascular aging, but the discussion should emphasize how these genetic markers can be translated into clinical practice. For instance, the authors could elaborate on the potential for genotype-based supplementation of L-citrulline to improve vascular health in aging populations. Including more practical applications would make the study more impactful.

Response:

We strongly agree with the reviewer that the practical implications of the genetic associations in this study need to be considered. As you indicated, the results of this study support the validity of genotype-based Cit supplementation in the future, which might make personalized supplements beneficial. We have added a statement including the possibility of further research development in the final paragraph of the Discussion section (page 20-21, lines 472–476, 480, and 482–486).

Comment from Reviewer #1

The limitation section is comprehensive, but it would be helpful to discuss potential future directions, such as validating these findings in other populations.

AND,

Comment from Reviewer #2

2. Broader Applicability of the Findings

The study focuses exclusively on the Japanese population. It would be beneficial to discuss the generalizability of these findings to other ethnic groups. The authors could also mention the importance of validating these results in diverse populations to ensure broader applicability.

Response:

Thank you for your comments. As you indicated, future validation in different populations and races is needed to better ensure the association between the SNPs found in this study and the symptoms of vascular aging. We have added this point in the seventh paragraph of the Discussion section (page 20, lines 449–453). With the addition of this description, paper #55, which indicates that GWAS results are not always replicated in other populations or races, was newly added to the reference list.

Comment from Reviewer #2

3. GWAS Results Interpretation

The manuscript includes the genome-wide association study (GWAS) findings, but the interpretation of the identified SNPs could be more detailed. The authors should clarify the functional relevance of these SNPs and the way they impact downstream nitric oxide signaling, particularly for the newly identified associations.

Response:

Thank you for pointing this out. The interpretation of the SNPs identified in the GWAS is provided in paragraph 6 of the Discussion section. We have already discussed PRMT6 in depth for its involvement with the NO signaling pathway. In response to your suggestion, we have also added the gene for the other identified SNP (HSD17B11) on page 19, lines 439 to 444, regarding its possible association with vascular aging. With the addition of this description, paper #54, which indicates that gene expression of HSD17B11 was significantly reduced by nitroglycerin, known to have vasodilating properties, was newly added to the reference list. However, as mentioned on the same page in lines 444 to 446, no association with the Cit and NO action pathways has been reported.

Comment from Reviewer #2

4. Methodological Details

While the methodology is described in detail, some sections could benefit from additional clarity. For example, the selection criteria for the traits used in association analyses should be more explicitly explained. Additionally, the rationale behind using both the candidate gene and GWAS approaches should be stated more clearly.

Response:

Thank you for this suggestion. I will first reply to the first half of your point. We used POMS2 and SF-36v2 and our visual analog scale (VAS) to determine the trait of association analysis in this study. We have added the detailed reasons for selecting each investigation method and the case/control selection criteria in the Trait definition section on page 6-8 to make it clearer. We will then respond to the latter part of your comment. Although candidate gene analysis has high power and is useful for finding associations between SNPs in specific genes and symptoms, it may miss other important SNPs. Therefore, we also performed GWAS to more broadly analyze SNPs across genes related to Cit/NO metabolism and the trait. We have added the rationale for using the candidate gene and the GWAS approach in the sixth paragraph of the Discussion section (page 18, lines 408–410).

Comment from Reviewer #2

Title

The title is informative but could be refined for clarity. Consider simplifying it to “SNP Associations in the L-Citrulline Pathway and Vascular Aging in the Japanese Population.”

Response:

Thank you for pointing this out. We have revised the title based on your suggestion to make it simpler.

Comment from Reviewer #2

Abstract

The abstract is comprehensive but slightly dense. Simplifying some of the terminology for a broader audience would improve readability. Additionally, the abstract should briefly highlight the practical implications of the study findings.

Response:

Thank you for pointing this out. We have simplified the sentences and phrases to make the Abstract more readable and added a note of the practical implications of the study findings at the end.

Comment from Reviewer #2

Figures and Tables

The figures and tables are well-organized, but the legends could be more descriptive. Providing additional context for each figure would help readers better understand the results at a glance.

Response:

In response to your suggestion, we have added background explanations to the legend of each figure for better understanding by the reader. In Fig 1 legend, we have added a brief description of the main purpose of the figure and the results. In Fig 2 legend, we have added a description of the role of the genes encoded by SNPs observed to be associated with the Cit and NO pathways in our study.

Comment from Reviewer #2

Terminology Consistency

Ensure consistency in using abbreviations and technical terms throughout the manuscript. For example, "L-citrulline" should be consistently abbreviated as "Cit" after its first use.

Response:

Thank you very much for your direction. Again, we have checked the consistency of terminology throughout the manuscript and corrected the inadequacies.

---

## [Editor Report · Decision Letter 2]

15 Apr 2025

SNP associations in the L-citrulline metabolic pathway and vascular aging in the Japanese population

PONE-D-24-52040R2

Dear Dr. Morita,

We’re pleased to inform you that your manuscript has been judged scientifically suitable for publication and will be formally accepted for publication once it meets all outstanding technical requirements.

Kind regards,

Asif Jan, Ph.D

Academic Editor

PLOS ONE
---

## [Editor Report · Acceptance letter]

PONE-D-24-52040R2

PLOS ONE

Dear Dr. Morita,

I'm pleased to inform you that your manuscript has been deemed suitable for publication in PLOS ONE. Congratulations! Your manuscript is now being handed over to our production team.

Kind regards,

on behalf of

Dr. Asif Jan

Academic Editor

PLOS ONE